# The Genomic Landscape of Urothelial Carcinoma with High and Low *ERBB2* Expression

**DOI:** 10.3390/cancers15245721

**Published:** 2023-12-06

**Authors:** Agreen Hadadi, Harris B. Krause, Andrew Elliott, Jacqueline T. Brown, Bassel Nazha, Lara R. Harik, Bradley C. Carthon, Benjamin Miron, Chadi Nabhan, Pedro C. Barata, Mohamed Saleh, Yuanquan Yang, Rana R. McKay, Mehmet A. Bilen

**Affiliations:** 1Department of Hematology and Medical Oncology, Emory University School of Medicine, Atlanta, GA 30322, USA; ahadadi@emory.edu (A.H.); jacqueline.theresa.brown@emory.edu (J.T.B.); bassel.nazha@emory.edu (B.N.); bradley.c.carthon@emory.edu (B.C.C.); 2CARIS Life Sciences, Inc., Irving, TX 75039, USA; hkrause@carisls.com (H.B.K.); aelliott@carisls.com (A.E.); cnabhan@carisls.com (C.N.); 3Department of Pathology and Laboratory Medicine, Emory University School of Medicine, Atlanta, GA 30322, USA; lara.harik@emory.edu; 4Department of Hematology and Oncology, Fox Chase Cancer Center, Philadelphia, PA 19111, USA; benjamin.miron@tuhs.temple.edu; 5Section of Hematology and Medical Oncology, Deming Department of Medicine, Tulane University School of Medicine, New Orleans, LA 70112, USA; pedro.barata@uhhospitals.org; 6University Hospital Seidman Cancer Center, Cleveland, OH 44106, USA; 7The Ohio State University Comprehensive Cancer Center, Columbus, OH 43210, USA; mohamed.saleh@osumc.edu (M.S.); yuanquan.yang@osumc.edu (Y.Y.); 8University of California San Diego, La Jolla, CA 92093, USA; rmckay@health.ucsd.edu

**Keywords:** *ERBB2*, HER2, RNA-seq, urothelial carcinoma, genomic landscape, precision oncology, tumor immune microenvironment

## Abstract

**Simple Summary:**

The present study demonstrates a high concordance between the immunohistochemistry (IHC) expression of HER2 and the RNA expression of *ERBB2* in urothelial carcinoma. The genomic, transcriptomic, and immunologic landscapes of urothelial carcinoma based on *ERBB2* expression were investigated. High-expressing *ERBB2* tumors have an increased expression of antibody drug conjugate (ADC) target genes *NECTIN4* and *TACSTD2* versus the low-expressing *ERBB2* tumor. Targeting *ERBB2*-high tumors with multiple ADCs should be investigated as a potential therapeutic strategy. Additionally, a positive association between *ERBB2* expression and survival was observed and warrants further investigation.

**Abstract:**

Background: Recent data suggests that HER2-targeted treatment is efficacious in urothelial carcinoma (UC). We investigated the genomic, transcriptomic, and immune landscapes and clinical outcomes in UC segmented by *ERBB2* expression. Methods: NextGen DNA/RNA sequencing was performed for 4743 UC tumors. A total of 3% (124/4125) of tumors had HER2 IHC and whole transcriptome sequencing (WTS) data. *ERRB2*-high and -low tumors were defined by ≥75th and <25th percentiles of *ERBB2* expression, respectively. PD-L1 (SP142) positive staining was defined as ≥2+ and ≥5%. HER2 (4B5) positive staining was defined as ≥3+ and >10% or 2+ and >10% with positive HER2 in situ hybridization (ISH). Results: Of the patients who were *ERBB2*-high, 79% (61/77) were HER2 positive via IHC. Tumors from lower tract UC had higher *ERBB2* expression compared to upper tract UC (50 v 40 median TPM (mTPM), *p* < 0.001). *ERBB2* expression was similar between primary and metastatic tumors (47 v 47 mTPM, *p* = 0.95). *ERBB2*-high tumors had a higher prevalence of pathogenic mutations in *pTERT*, *ERBB2*, and *ELF3* versus *ERBB2*-low tumors, *p* < 0.001. *ERBB2*-high tumors had higher expressions of ADC target genes *NECTIN4* (12 v 8 mTPM) and *TACSTD2* (366 v 74 mTPM) versus *ERBB2*-low (*p* < 0.001), as well as better overall survival from time of tissue sampling than *ERBB2*-low (HR 1.71, *p* < 0.001). Conclusion: Our study demonstrated a high concordance between HER2 expression by IHC and *ERBB2* gene expression by WTS in UC. Differences in ADC target expression between *ERBB2*-high vs. *ERBB2*-low UC may provide a rationale for combination treatment strategies with HER2-ADC. The association between high *ERBB2* expression and survival advantage warrants further investigation.

## 1. Introduction

Bladder cancer is the fourth most common cancer in men [1]. Its incidence continues to rise, with an expected 82,290 new cases in 2023 [1]. Urothelial carcinoma (UC) comprises most urinary bladder cancers (>95%) and can be further classified as muscle invasive or non-muscle invasive (MIBC or NMIBC, respectively). Among patients with UC, approximately 20% of cases are muscle invasive at time of diagnosis, with a higher likelihood of metastasis or recurrence following radical surgical resection and a significantly lower 5-year survival rate [2]. Currently, outcomes are poor in patients with advanced UC treated with first- and second-line systemic therapy, specifically platinum-based therapies, immune checkpoint inhibitors (i.e., pembrolizumab), more recently, tyrosine kinase inhibitors (erdafitinib) [3], and antibody–drug conjugates (ADC) (i.e., enfortumab vedotin and sacituzumab govitecan) [4,5]. Because of this, there is a need for investigating novel therapies in advanced UC.

The HER2 protein, encoded by the *ERBB2* gene, is a receptor tyrosine kinase that is a part of the epidermal growth factor receptor family [2]. Expression of HER2 via immunohistochemistry (IHC) occurs in a subset of urothelial cancers, more frequently in MIBC and metastatic bladder cancer than in NMIBC, and is linked to *ERBB2* gene amplification in a subset of patients [6,7]. Although HER2 overexpression is associated with tumorigenesis and cell proliferation in several cancers, including UC [8], the prognostic capabilities of HER2 expression in UC remain an important unanswered question. A preponderance of studies suggests that HER2 expression is associated with poor prognosis in UC [8,9,10]. However, some studies suggest a positive association between HER2 expression and survival, although this is less conclusive, with a smaller patient cohort [9].

HER2 expression and *ERBB2* amplification have been studied as biomarkers that can predict sensitivity to HER2-targeted monoclonal antibodies and ADCs [10,11]. Recent data demonstrate that treatment with HER2-targeted ADCs can lead to improved clinical outcomes, specifically progression-free survival and overall survival in HER2-expressing UC [12]. With this emerging data, HER2-targeted therapies are being tested in patients with bladder cancer harboring HER2 overexpression and/or amplification of *ERBB2*.

With the development of novel HER2-targeted therapies, there is a need to accurately define HER2 expression. To date, there are no standardized UC criteria to assess HER2 status. As a result, clinicians must rely on HER2 IHC staining guidelines for breast cancer and gastric cancer, which can reduce accuracy [11]. With regards to breast cancer, in the current standard of practice, patients with a +3 score are directly enrolled into trastuzumab therapy, while those with a +2 score are recommended to undergo additional ISH methods for gene amplification detection. The +1 HER2-positive cases are not considered positive, with no recommendation to verify via ISH [11]. Monoclonal antibodies, TKIs, and ADCs are novel HER2-targeted therapies whose efficacies have been investigated based on HER2 expression via IHC [13].

In this study, we describe the concordance between *ERBB2* vs. HER2 expression and define the genomic, transcriptomic, and immunological landscapes of *ERBB2*-high- and -low-expressing UC tumors (via RNA expression) using a large database of real-world patient samples. We hypothesized that there would be high concordance between *ERBB2*/HER2 expression across whole transcriptome sequencing (WTS), copy number amplification (CNA), and IHC.

## 2. Methods

### 2.1. Study Cohort

Formalin-fixed paraffin-embedded (FFPE) patient samples (*n* = 4743) were submitted to a commercial CLIA-certified laboratory (Caris Life Sciences, Phoenix, AZ, USA). For inclusion in this study, the tumor must have undergone either a targeted 592-gene panel, whole-exome sequencing (WES), or whole-transcriptome sequencing (WTS). In total, 124 (3.0%) tumors were analyzed by IHC for HER2 expression and WTS, 4125 (87%) tumors were analyzed by WTS, and 4610 (97%) were analyzed by NGS/WES. The datasets analyzed are not publicly available but are available from the corresponding author upon reasonable request. The present study was conducted in accordance with guidelines of the Declaration of Helsinki, Belmont Report, and U.S. Common Rule. With compliance to policy 45 CFR 46.101(b), this study was conducted using retrospective, de-identified clinical data, and patient consent was not required.

### 2.2. Defining Primary, Metastatic, Lower, and Upper Urothelial Tract UC

A primary (local) tumor was defined as anything where the primary site was labeled as the bladder, and specimen sites were labeled as the bladder or renal/kidney. Lower and upper urothelial tracts (LTUC and UTUC, respectively), as well as primary versus metastatic sites were defined by the annotated primary site and specimen (biopsy) site. UTUC corresponded to tumors arising in the kidney and ureter, while LTUC referred to tumors arising in the urinary bladder and/or urethra. A metastatic (non-local) tumor was defined as any non-primary tumor.

### 2.3. DNA Next-Generation Sequencing (NGS)

A targeted 592-gene panel or WES was performed using genomic DNA isolated from FFPE tumor samples. The 592-gene panel was sequenced using the NextSeq platform (Illumina, Inc., San Diego, CA, USA). A custom-designed SureSelect XT assay was used to enrich 592 whole-gene targets (Agilent Technologies, Santa Clara, CA, USA). NGS detected variants with >99% confidence based on allele frequency and amplicon coverage, with an average sequencing depth of coverage of >500 and an analytic sensitivity of 5%.

WES was performed using the Illumina NovaSeq 6000 sequencers (Illumina, Inc.). A hybrid pull-down panel of baits designed to enrich more than 700 clinically relevant genes at high coverage and high read-depth was used. A 500 Mb SNP backbone panel (Agilent Technologies) was added to assist with gene amplification measurements. The performance of the WES assay was validated for sequencing variants, copy number alteration, tumor mutational burden, and micro-satellite instability. The test was validated to 50 ng of input and had a PPV of 0.99 against a previously validated NGS assay. WES can detect variants with tumor nuclei as low as 20% and will detect variants down to 5% variant frequency, with an average depth of at least 500×. Matched normal tissue was not sequenced.

### 2.4. Identification of Genetic Variants and Copy Number Amplification (CNA)

Genetic variants identified were interpreted by board-certified molecular geneticists and categorized as ‘pathogenic’, ‘likely pathogenic’, ‘variant of unknown significance’, ‘likely benign’, or ‘benign’, according to the American College of Medical Genetics and Genomics (ACMG) standards. When assessing mutation frequencies of individual genes, ’pathogenic’, and ‘likely pathogenic’ were counted as mutations, while ‘benign’, ‘likely benign’ variants, and ‘variants of unknown significance’ were excluded. The CNA of each exon was determined by calculating the average depth of the sample along with the sequencing depth of each exon and comparing this calculated result with a pre-calibrated value.

### 2.5. Tumor Mutational Burden (TMB)

TMB was measured by counting all non-synonymous missense, nonsense, inframe insertion/deletion, and frameshift mutations found per tumor that had not been previously described as germline alterations in dbSNP151 and Genome Aggregation Database (gnomAD) databases or benign variants identified by Caris geneticists. A cutoff point of ≥10 mutations per MB was used, based on the KEYNOTE-158 pembrolizumab trial [14], which showed that patients with a TMB of ≥10 mt/MB across several tumor types had higher response rates than patients with a TMB of <10 mt/MB. Caris Life Sciences is a participant in the Friends of Cancer Research TMB Harmonization Project [15].

### 2.6. MSI/MMR Status

A combination of two test platforms was used to determine the MSI or MMR status of the tumors profiled, including IHC (MLH1, M1 antibody; MSH2, G2191129 antibody; MSH6, 44 anti-body; and PMS2, EPR3947 antibody [Ventana Medical Systems, Inc., Tucson, AZ, USA]) and NGS (>2800 target microsatellite loci were examined and compared to the reference genome hg19 from the University of California, Santa Cruz Genome Browser database). The two platforms generated highly concordant results, as previously reported [16], and in the rare cases of discordant results, the MSI or MMR status of the tumor was determined in the order of IHC followed by NGS.

### 2.7. Whole Transcriptome Sequencing

FFPE specimens underwent pathology review to diagnose percent tumor content; a minimum of 10% of tumor content was required to enable the extraction of tumor-specific RNA. The Qiagen RNA FFPE tissue extraction kit was used for extraction, and the RNA quality and quantity were determined using the Agilent TapeStation. Biotinylated RNA baits were hybridized to the synthesized and purified cDNA targets, and the bait–target complexes were amplified in a post-capture PCR reaction. The resultant libraries were quantified and normalized, and the pooled libraries were denatured, diluted, and sequenced; the reference genome used was GRCh37/hg19, and analytical validation of this test demonstrated ≥97% positive percent agreement (PPA), ≥99% negative percent agreement (NPA), and ≥99% overall percent agreement (OPA) with a validated comparator method. For transcript counting, transcripts per million molecules were generated using the Salmon expression pipeline. *ERBB2*-high and -low expressions were defined as ≥top and <bottom quartiles of *ERBB2* transcripts per million (TPM), respectively.

### 2.8. RNA Signatures

WTS data was used to calculate the immune cell fraction using quanTIseq immune deconvolution [17]. QuanTIseq is an immune deconvolution algorithm that utilizes RNA transcripts that are known to be expressed in specific immune cell types to deconvolute bulk RNA sequencing data and to predict the different immune cell fractions that are present in the bulk RNA lysate. WTS data was also used to calculate a T-cell inflamed score, as previously described [18].

### 2.9. Immunohistochemistry (IHC) for PD-L1 and HER2

IHC was performed on FFPE sections of glass slides. Slides were stained using automated staining techniques, per the manufacturer’s instructions, and were optimized and validated per CLIA/CAP and ISO requirements. Staining was scored for intensity (0 = no staining; 1+ = weak staining; 2+ = moderate staining; and 3+ = strong staining) and staining percentage (0–100%). Results were categorized as positive or negative by defined thresholds specific to each marker, based on published clinical literature that associates biomarker status with patient responses to therapeutic agents. PD-L1 (SP142) positive staining was defined as ≥2+ and ≥5%, HER2 (4B5) high positive staining was defined as ≥3+ and >10% or 2+ and >10% with positive HER2 ISH (in situ hybridization), HER2 low was defined as either 2+ and >10% with negative HER2 ISH or 1+ and >10%, and negative staining was defined as either <10% of cells having staining or tumors with 0+ staining intensity.

### 2.10. Clinical Outcomes

Real-world overall survival was obtained from insurance claims and calculated from either tissue collection or from the start of the immune checkpoint inhibitor (ICI: atezolizumab, avelumab, nivolumab, or pembrolizumab) to the last contact. Kaplan–Meier estimates were calculated for molecularly defined patients.

### 2.11. Statistics

Descriptive statistics were used to summarize and compare molecular features. Mann–Whitney U, Kruskal–Wallis, and X^2^/Fisher Exact tests were applied where appropriate, with *p*-values adjusted for multiple comparisons (*p* < 0.05).

## 3. Results

### 3.1. Cohort Description

Among 4743 UC samples, 72% were male, and the median age of the cohort was 72 years. A total of 65% of the tumors were from the primary site (bladder and upper tract), and 35% were from metastatic sites. Median *ERBB2* expression was similar between primary and metastatic tumors (47 v 47 TPM, *p* = 0.95) but significantly higher in liver metastases (59 v 45 TPM, *p* < 0.001) (Figure 1A). Of the 3999 with a LTUC or UTUC designation, 19.9% (795/3999) were UTUC. LTUC tumors had a higher median *ERBB2* expression, compared to UTUC (50 v 40 TPM, *p* < 0.001) (Figure 1B). A total of 40.1% of UTUCs came from a metastatic site, as compared to 32.3% of LTUCs, 23.7% (*p* < 0.001). There was no difference in the distributions of the metastatic sites between metastatic UTUC vs. LTUC (13.7% vs. 11.4% liver, *p* = 0.16 and 13.0% vs. 11.3% lung, *p* = 0.31)

### 3.2. Concordance Analysis

Concordance was determined for *ERBB2* expression by *ERBB2* WTS vs. HER2 IHC and *ERBB2* WTS vs. *ERBB2* CNA. Of the tumors that had HER2 IHC and *ERBB2* WTS data available, 86/198 (43%) of tumors were HER2 positive by IHC, 51/198 (26%) were HER2 low, and 61/198 (31%) were HER2 negative. Totals of 94% of HER2 positive, 81% of HER2 low, and 95% of HER2 negative tumors were LTUC (*p* = 0.08). Totals of 59% of HER2-positive, 45% of HER2-low, and 24% of HER2-negative tumors were from the primary site (*p* = 0.02).

HER2-positive tumors by IHC had a significantly higher median expression of *ERBB2,* as compared to HER2-low and HER2-negative (449 vs. 68 vs. 27 TPM, *p* < 0.001) (Figure 2). Next, the concordance of ERRB2 expression (divided into quartiles by expression) was compared to HER2 staining (positive vs. low vs. negative). Of the patients who were in the top expression quartile of *ERBB2* expression, 79% (61/77) were HER2-positive, 16% (12/77) were HER2-low, and 5% (4/77) were HER2-negative. Of those tumors that were in the bottom quartile of *ERBB2* expression, 15% (2/13) were HER2-positive, 31% (4/13) were HER2-low, and 54% (7/13) were HER2-negative. Of the tumors that had HER2 IHC and *ERBB2* CNA data available, 97% (61/63) of tumors with high amplification (CNA > 6) had positive HER2 staining, while 5% (4/83) of those with no amplification (<4 CNA) had positive HER2 staining (Figure 3, Table 1 and Table 2).

### 3.3. Gene Alterations in ERBB2-High vs. ERBB2-Low Tumors

*ERBB2* expression was subdivided into four quartiles based on expression (Table 3). *ERRB2*-high and -low tumors were defined by ≥75th and <25th percentiles of *ERBB2* expression, respectively. *ERRB2* (12% vs. 5%), *pTERT* (76% vs. 60%), and *ELF3* (14% vs. 5%) had a higher prevalence of mutations in *ERBB2*-high compared to *ERBB2*-low tumors. The opposite pattern was observed for *CDKN2A* (4% vs. 9%), *KMT2D* (20% vs. 35%), and *PIK3CA* (13% vs. 23%), with a lower mutation prevalence in the *ERBB2*-high vs. *ERBB2*-low tumors, all *p* < 0.001 (Figure 3). The most common *ERBB2* pathogenic amino acid changes observed in the *ERBB2*-high group were 42% S310F (54/130), 11% S310Y (14/130), 7% V777L (9/130), and 7% I767M (9/130).

### 3.4. Immune Landscape and Markers of ICI Response in ERBB2-High vs. ERBB2-Low Tumors

Immune infiltrates inferred from RNA expression using quanTIseq immune deconvolution were assessed for *ERBB2*-low and -high tumors. When comparing the median percent immune infiltrate between *ERBB2*-high vs. *ERBB2*-low tumors, more neutrophil (9.2% vs. 6.7%) and M2 macrophages (6.2% vs. 4.8%) were present in the *ERBB2*-high group, while less M1 macrophages (1.1% vs. 3.5%) were observed in the *ERBB2*-high group (*p* < 0.05 for all) (Figure 4A,B). Additionally, the T-cell inflamed score, a transcriptional signature predictive of a response to ICI, was assessed across all quartiles with increasing amounts of intermediate tumors (responsiveness to ICI is unclear) as the expression increased and with little change in the proportion of inflamed tumors (more responsive to ICI) (Figure 4C). Of note, *ERBB2*-low tumors had a higher rate of PD-L1+ (by IHC), compared to *ERBB2*-high tumors (40.3% vs. 18%, *p* < 0.001) (Figure 4D). Finally, dMMR/MSI-H prevalence was slightly higher in *ERBB2*-low compared to *ERBB2*-high tumors (2.1% vs. 1.0%, *p* < 0.05). However, *ERBB2*-high tumors were more frequently TMB-high, as compared to *ERBB2*-low tumors (35% vs. 5.4%, *p* < 0.001).

### 3.5. Expression of ADC Targets in ERBB2-High vs. ERBB2-Low Tumors

*ERBB2*-high tumors had higher median expressions of ADC target genes *NECTIN4* (12 v 8 TPM, *p* < 0.001, Pearson correlation coefficient = 0.18) and *TACSTD2* (366 v 74 TPM, *p* < 0.001, Pearson correlation coefficient = 0.57), as compared to *ERBB2*-low tumors (Figure 5A,B).

### 3.6. Clinical Outcomes

*ERBB2*-high tumors were associated with significantly better overall survival (OS) from time of tissue sampling than those with *ERBB2*-low tumors (HR 1.34, 95% CI 1.14–1.60, *p* < 0.001), regardless of treatment regimen (platinum-based chemotherapy and/or ICI) (Figure 5C). Additionally, *ERBB2*-high tumors were associated with better OS from the start of ICI (HR 1.51, 95% CI 0.938–1.412, and *p* = 0.178) (Figure 5D, Appendix A).

## 4. Discussion

Our study demonstrates a high concordance between HER2 expression by IHC and *ERBB2* gene expression by WTS or *ERBB2* copy number amplification in UC. As NGS tumor profiling becomes more widely available, RNA expression of *ERBB2* could be a potential surrogate for HER2 status. To date, there is little data comparing *ERBB2* expression, HER2 CNA, and HER2 IHC. Additionally, by demonstrating the concordance of IHC and WTS, we were able to greatly expand our cohort for the subsequent analysis, as there were over an order of magnitude more cases sequenced by WTS than IHC in our cohort.

Urothelial cancer has notable genomic heterogeneity [8], with mutations and amplifications of *ERBB2* in 6–17% of samples [11]. Large-scale genomic profiling via The Cancer Genome Atlas (TGCA) has demonstrated ERBB2 amplification in UC [19]. This was further validated in the context of the tumor molecular subtype via the Lund taxonomy classification system, which demonstrated ERBB2 amplification in the genomic unstable (GU) subtype, also considered the luminal subtype, of UC [20], and absent ERBB2/ERBB3 expression in the basal/squamous cell carcinoma (SCC)-like subtype [21]. NECTIN-4 expression was also found to be significantly elevated in the luminal subtype [22]. In our sample cohort, *ERBB2*-high-expressing tumors had higher expressions of ADC target genes *NECTIN4* and *TACSTD2*. This suggests that additional ADC therapies could target ERBB2-high tumors either simultaneously or in sequence to reduce the incidence of therapy resistance. Understanding UC classification in terms of phenotypic subtype, ERBB2 expression, and coexisting mutations is crucial to gain a better grasp on and exploit potentially targetable mutations that could be used alongside HER2-targeting ADC in HER2-positive UC.

In our cohort, ERBB2-high tumors had a higher prevalence of pathogenic ERBB2 mutations. ERBB2-ADC binds to the extracellular domain of the HER2, preventing activation through ligand binding, increasing internalization of the HER2 in addition to the delivery of a toxic payload. HER2-ADC may be modulated by these pathogenic ERBB2 mutations. Interestingly, patients with HER2-positive breast cancer with ERBB2 mutations are less responsive to trastuzumab than ERBB2 wild-type patients [23]. Alternatively, a case report on an *ERBB2* mutant breast tumor (S310F) that was HER2 nonamplified experienced a prolonged response to trastuzumab in combination with pertuzumab and fulvestrant [24]. Of note, S310F was the most common codon mutation observed amongst ERBB2 mutants in our cohort. The prevalence of ERBB2 mutations in tumors likely eligible for HER2-ADC highlights the importance of investigating how *ERBB2* mutation status affects HER2-ADC efficacy. Future works should try to identify specific mutations that could be used as biomarkers of response to HER2-ADC.

Additionally, we observed an increased prevalence of *pTERT* mutations in the ERBB2-high cohort. As therapies that target *pTERT* are developed, co-administering HER2-ADC and *pTERT*-targeting drugs could be an efficacious strategy in the future [25].

Aside from a lack of standardization of HER2 testing, another major barrier to the success of HER2-targeted therapy is the lack of information regarding the co-expression of immunomodulatory biomarkers [8]. With ongoing clinical trials exploring the use of ICI in UC [26,27], it is important to better characterize the immune landscape of *ERBB2*-high- and -low-expressing UC tumors. *ERBB2*-low tumors had a higher rate of PDL1+ staining, as compared to *ERBB2*-high tumors. Since there are currently limited treatment options for HER2-low UC, this finding may offer insight into directing more attention towards ICI as a potential treatment, monotherapy or combined, for this patient population.

Our study showed that OS was longer in the *ERBB2*-high cohort. This is interesting, as most data looking at *ERBB2*/HER2 status and clinical outcomes have shown that HER2 positivity is associated with high tumor grade, lymph node metastasis, and UC-specific mortality [9,28]. It should be emphasized that analysis of OS was markedly limited by unmeasured confounders, including but not limited to performance status, medical comorbidities, organ function, prior surgery or therapeutics administered, site(s) of metastasis, and systemic therapies administered. As a result of these significant limitations, future work should aim to verify the association between ERBB2 expression and clinical outcomes using multivariate analysis to account for clinical and molecular confounders.

In a recent meta-analysis by Gan et al. that evaluated the prognostic value of HER2 in bladder cancer, it was reported that the HER2 gene was not associated with OS, disease-specific survival, or progression-free survival, but it was related to advanced pathological stage, high tumor grade, and tumor recurrence [10]. It is well established that HER2 status is associated with improved outcomes in patients treated with cytotoxic chemotherapy in breast cancer and hypothesized that this may reflect these patients having a better baseline functional status, since that is required to receive platinum-based therapy, given the high side-effect profile [9]. In a more recent study investigating the prognostic value of HER2 expression in UTUC, patients with HER2 overexpression, defined as 3+ staining by IHC, were noted to have a longer median OS (60 months), as compared to patients with a negative HER2 expression who had a median OS of 27.5 months [28]. The implications of improved clinical outcomes associated with higher *ERBB2* expression needs to be further explored.

The strengths of this study include a large sample size of over 4000 patient samples and that all NGS testing and IHC staining were completed by one provider, Caris, which reduced variability across methodologies used for *ERBB2*/HER2 quantification. Furthermore, this study exclusively utilized retrospective data, and not every patient who underwent genomic sequencing also underwent HER2 IHC staining. Due to the limited number of tumors stained for HER2, *ERBB2*-high expression was defined as Q4 of ERBB2-expressing tumors via WTS, while *ERBB2*-low expression was defined as Q1 of ERBB2-expressing tumors. Of the tumors that had both HER2 IHC and WTS available, ERBB2-high was concordant with HER2-positive tumors, and ERBB2-low was concordant with HER2-negative tumors. Additional limitations include the lack of matched/paired non-cancer tissue and the lack of germ-line testing to identify patients with Lynch syndrome as well as tumor heterogeneity, which can contribute to the variability of HER2 status across tumor types. Finally, tumors that undergo NGS tend to be recurrent or of an advanced stage and have often undergone treatment prior to sequencing, which creates a selection bias. For these reasons, our data set may not be representative of all UC.

## 5. Conclusions

Our study explored the role of *ERBB2*/HER2 expression in UC with regards to the genomic landscape, tumor immune landscape, and clinical outcomes in a large database of real-world patient samples. *ERBB2*-high tumors had significantly more pathogenic ERBB2 mutations, and the role ERBB2 mutations play in modulating HER2-ADC efficacy should be further explored. The high expressions of ADC target genes *NECTIN4* and *TACSTD2* were observed. Targeting *ERBB2*-high tumors with multiple ADCs should be investigated as a potential therapeutic strategy. A positive association between *ERBB2* expression and survival was observed and warrants further investigation.

## Figures and Tables

**Figure 1 cancers-15-05721-f001:**
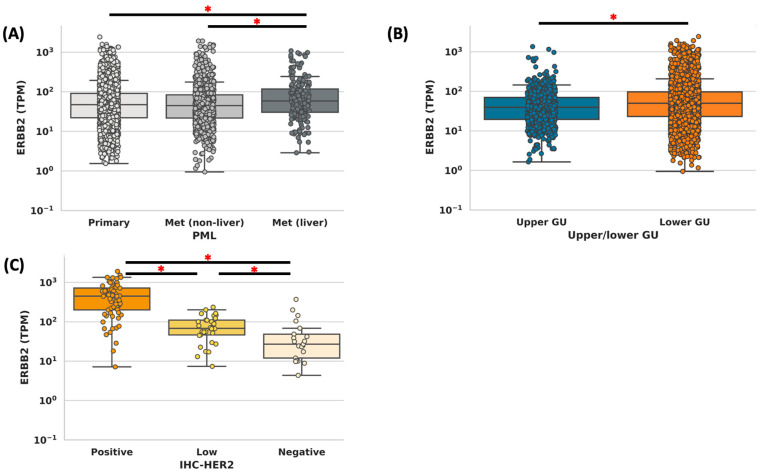
(**A**) Expression of *ERBB2* for tumors that were collected from primary, metastatic (all sites but liver), or liver sites. (**B**) Expression of *ERBB2* for upper and lower tract tumors. (**C**) Expression of *ERBB2* in transcripts per million (TPM) for tumors that were IHC HER2 positive, low, and negative (asterisk indicates significance, *p* < 0.05).

**Figure 2 cancers-15-05721-f002:**
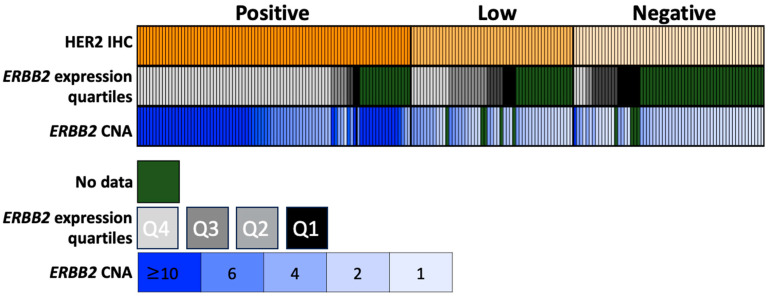
Concordance of IHC staining, *ERRB2* expression by quartile, and *ERBB2* CNA.

**Figure 3 cancers-15-05721-f003:**
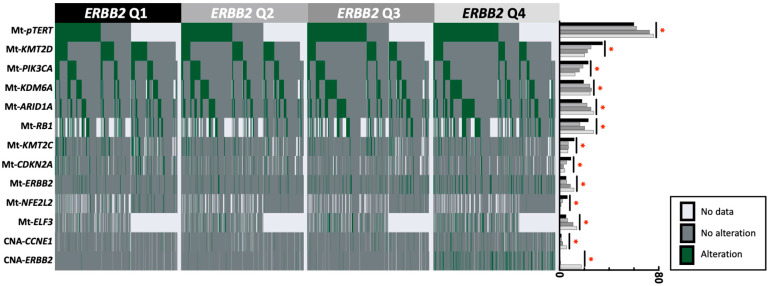
Oncoprint and prevalence of selected pathogenic alterations in urothelial carcinoma tumors associated with the high or low expression of *ERBB2* (genes shown have >4% difference in prevalence between *ERBB2* Q4 and Q1). * indicates statistical significance.

**Figure 4 cancers-15-05721-f004:**
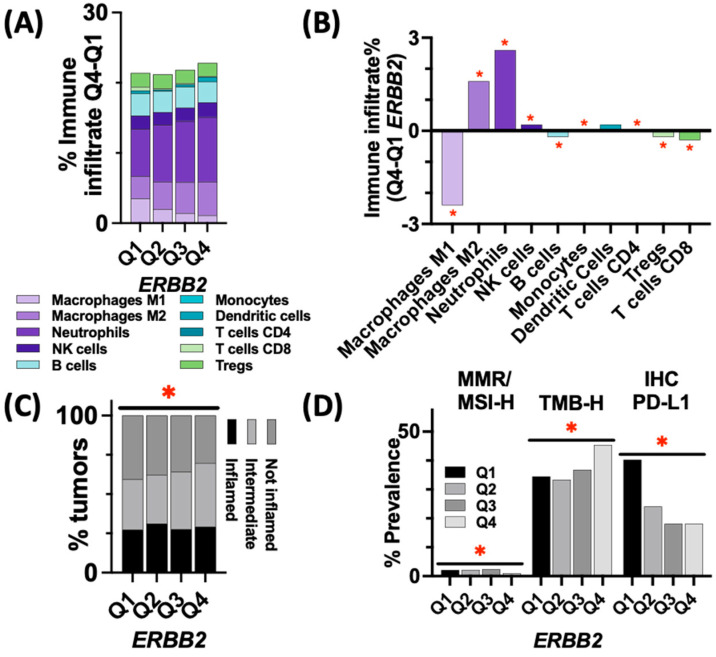
(**A**) Immune infiltrates for *ERBB2*-expressing tumors (inferred from RNA expression using quanTIseq immune deconvolution) and (**B**) the differences in immune infiltrate between *ERBB2* expression quartiles 4 and 1 (*p* < 0.05). (**C**) T cell-inflamed score across *ERBB2* expression quartiles (*p* < 0.001). (**D**) Prevalence of immune biomarkers based on *ERBB2* expression (* indicates *p* < 0.05).

**Figure 5 cancers-15-05721-f005:**
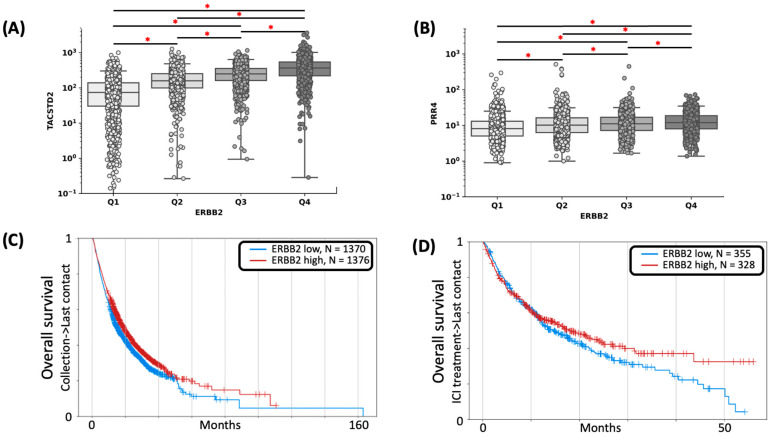
Expressions of *TACSTD2* (**A**) and *PRR4* (**B**) across *ERBB2* expression quartiles (*p* < 0.05). (**C**) Overall survival (OS) for *ERBB2*-high vs. *ERBB2*-low tumors (HR 1.71, 95% CI 1.065–1.286, and *p* < 0.001). (**D**) OS since treatment with ICI (HR 1.51, 95% CI 0.938–1.412, and *p* = 0.178). (* indicates *p* < 0.05).

**Table 1 cancers-15-05721-t001:** Concordance between IHC and WTS.

		WTS			
		Q1	Q2	Q3	Q4
IHC	Positive	2	2	5	61
		15% (2/13)	13% (2/15)	26% (5/19)	79% (61/77)
	Low	4	5	12	12
		31% (4/13)	33% (5/15)	63% (12/19)	16% (12/77)
	Negative	7	8	2	4
		54% (7/13)	54% (8/15)	11% (2/18)	5% (4/77)

**Table 2 cancers-15-05721-t002:** Concordance between IHC and CNA.

		CNA		
		Not Amplified	Intermediate	Amplified
		(<4)	(>4)	(>6)
IHC	Positive	4	21	61
		5% (4/83)	42% (21/42)	97% (61/63)
	Low	30	15	1
		36% (30/83)	42% (15/42)	1.5% (1/63)
	Negative	49	6	1
		59% (49/83)	16% (6/42)	1.5% (1/63)

**Table 3 cancers-15-05721-t003:** Cohort characteristics.

	*ERBB2* Q1	*ERBB2* Q2	*ERBB2* Q3	*ERBB2* Q4	Statistic	q-Value
Count (N)	1032	1031	1031	1031		
Median Age [range] (N)	72 [26–>89] (1032)	72 [18–>89] (1031)	72 [28–>89] (1031)	72 [24–>89] (1031)	Kruskal-Wallis	0.32
Female	34.50%	29.00%	25.60%	22.70%	chi-square	<0.001
	(356/1032)	(299/1031)	(264/1031)	(234/1031)		

## Data Availability

The data used in this study are not publicly available but can be made available upon reasonable request.

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
