# Peer review of "The Genomic Landscape of Urothelial Carcinoma with High and Low ERBB2 Expression"

_cancers, 2023, doi:10.3390/cancers15245721_

Round 1
Reviewer 1 Report
Comments and Suggestions for Authors
Overall Impression:
The paper looks closely at ERBB2 gene changes in bladder cancer and uses data from many patients. It’s a solid study, but the way patients were chosen could influence the results. Also, they may be too sure about ERBB2's effect on survival without considering all factors.
Abstract:
The abstract is clear and direct, giving a snapshot of the research, how it was done, the findings, and their potential meaning. It's straightforward and gives you the gist quickly.
Introduction:
The beginning explains why the HER2 and ERBB2 genes matter in bladder cancer. It makes a good point about needing a consistent way to measure HER2. Still, it could talk more about how these genes might predict patient outcomes.
Methods:
The study's steps are explained in detail, which is key for others to replicate the work. The criteria for including or not including patients are well thought out. Comparing cancerous to normal cells wasn't done but could give a better overall picture. Since the study looks back at old data, not needing patient permission is reasonable.
Results:
The results are full of detail, with helpful images and charts. The match between HER2 protein and ERBB2 gene levels stands out. But they should be more cautious in connecting these gene levels to patient survival, as there could be more to the story.
Discussion:
The end discussion places the results in the bigger picture of bladder cancer research. It argues well for considering ERBB2 in both diagnosis and treatment. Yet, it could talk more about the study's limits and how they might sway the results, like the connection between ERBB2 and how long patients live.
Reviewer 2 Report
Comments and Suggestions for Authors
Congratulations for your work.
Knowledge of genetical landscape of neoplasm is becoming a need unreponded at today. Integration of newel therapies as antibodies is driving clinicians to other paths encompassing sistemic therapies and surgeries.
limits of your paper are the origin of dataset, the publicity of data.
Potential relevant clinical data are missing on pathological state and other features potentially relevant in this landfield. Morever clinica data influencing OS are missing. So your conclusion on survival have to be balanced by a number of potential confounders. FInally missing data about any kind of therapy/surgery however potentially impacting your result and survival.
On this very impressive cohort we don't know how many patient could have diagnosed with Lynch syndrome and the relationship with pother genetical cascade, probably you could add this consideration.
However these data give an help in enlarging knowledge on UTUC pathogenic cascade but need a more clinically focused application.
In conclusion add hypotesis of more clinical applications, from diagnosis (urinary citologies or functional imaging) to therapy scenarios.
Reviewer 3 Report
Comments and Suggestions for Authors
This study described the role of ERBB2/HER2 expression in UC with regards to the genomic landscape, tumor immune landscape and clinical outcomes in a large patient's database. This is very comprehensive genomic and immunohistochemical research well supported bioinformatically. The most important conclusion is the relation of tumor status and scientific outcome with mutations and alterations in ErbB2 gene. Another important discovery is good correlation with gene expression and protein HER2 presence. This will allow to perform deep sequencing instead of more complicated and less effective immunohistochemistry. The last finding -correlation with ErBB2 and NECTIN 4 expression is also very critical because it will give the opportunity to make prediction for treatment with NECTIN4 targeting drugs.
